# The Impact of the Localization of Metastasis in Bladder Cancer Patients with Recurrence After Cystectomy

**DOI:** 10.3390/cancers17050867

**Published:** 2025-03-03

**Authors:** Mads Aamand, Simone Buchardt Brandt, Rikke Vilsbøll Milling, Jørgen Bjerggaard Jensen

**Affiliations:** 1Department of Urology, Aarhus University Hospital, 8200 Aarhus, Denmark; simbra@rm.dk (S.B.B.); rimill@rm.dk (R.V.M.); bjerggaard@skejby.rm.dk (J.B.J.); 2Department of Clinical Medicine, Aarhus University, 8000 Aarhus, Denmark

**Keywords:** bladder cancer, cystectomy, metastases, recurrence, survival

## Abstract

This study investigates the impact of the localization of metastasis in patients with bladder cancer that were treated with cystectomy. Not many studies have investigated this subject before in this patient group. We used a clinical quality database for this study. This study’s findings indicate that patients with bone metastases have worse prognosis than patients with lymph node metastases. The study’s finding also indicate that patients with organ metastases had worse prognosis than patients with lymph node metastases or local recurrence. Furthermore, we found that patients with multiple metastases have worse prognosis compared to patients with single recurrence.

## 1. Introduction

Bladder cancer (BC) represents a frequent cancer disease and is currently ranked the ninth most common human cancer worldwide [1]. Radical cystectomy is most often the primary treatment of muscle-invasive BC [2]. However, up to half of patients develop recurrent disease after cystectomy. Recurrence of BC is either local or distant, most often diagnosed as distant metastases, commonly in lymph nodes, lungs, liver, or bones [3]. A previous study concluded that stage-based protocol would be beneficial due to different recurrence patterns [4]. Another retrospective study evaluated 343 BC patients, of which 158 (46.1%) experienced recurrent disease, whereby the authors found survival differences after recurrence when comparing local recurrence and distant metastases in patients who were treated with curative intentions. This study shows a slightly better prognosis regarding patients with local recurrence vs. patients with distant metastases [5]. A study found that metastatic BC patients with bone, brain, liver, and lung metastases had worse overall and cancer-specific survival than patients who did not have the corresponding site of metastasis [6]. A study has investigated factors that increase the risk of specific sites of metastases [7].

Other studies investigating other cancer types for example breast cancer found an association between the location of metastases and patients’ survival outcomes [8]. However, the association between the location of recurrence and patients’ prognosis has been studied sparely in BC. Until now, the location of metastases’ potential impact on survival has been studied in 2256 patients with stage IV BC using the North American SEER database, with results indicating different survival outcomes depending on the site of metastases [9]. To the best of our knowledge, the specific localisation of recurrent disease after radical cystectomy has not been studied previously, and neither has its potential impact on patients’ survival.

The aim of this study was to investigate whether the specific localization of recurrence and the number of metastases had an impact on patients’ survival outcomes.

## 2. Materials and Methods

All patients diagnosed with BC who had undergone radical cystectomy with extended lymph node dissection and experienced BC recurrence between 2015 and 2021 at a single-tertiary referral centre were eligible for inclusion. All patients underwent follow-up, including a CT-TAP scan, at the department of urology after the primary surgery with 4-month, 1-year, and 2-year follow-up visits. High risk patients had additional 8-, 18-, and 36-month follow-up visits. All data were collected from a clinical quality database in the Midtjylland region using REDCap 14.5.36. Data were systematically collected from the electronic medical record. Data included the preoperative demographics, comorbidities, and pathological and clinical follow-up data. Treatment with neoadjuvant chemotherapy (NAC) was registered dichotomously (YES/NO). Recurrence was radiologically found and histologically confirmed in most cases. Cause of death was determined using a patient journal and was categorised as cancer-specific or other. In some cases, it was not possible to achieve biopsies due to ethical reasons. Additionally, using the research protocol, recurrence using circulating tumour DNA (ctDNA) was detected [10].

### Statistical Analysis

The follow-up time was calculated from recurrence to death or censoring. Log-rank tests were used to calculate the statistical difference in overall survival. When using Cox regression to examine the hazard ratio (HR), we assumed proportional hazards, using Schoenfeld residuals. The groups of interest were based on location of recurrence, such as local or lymph nodes in the lung, bone, and liver, as they were the most common sites of metastases in our study population. The group expected to have the highest survival probability served as the reference group. Adjusted HRs were calculated for all groups, adjusted for sex, age, neoadjuvant chemotherapy, ASA classification, and nodal status at time of cystectomy. All data analysis was performed in R studio version 5.0 [11].

## 3. Results

Between 2015 and 2021, 666 patients underwent radical cystectomy, of which 180 (27%) had recurrence and were included in this study. The cohort characteristics are illustrated in Table 1. Two patients were excluded due to having underwent cystectomy for benign reasons. In the Appendix A is provided, comparing the baseline characteristics of all patients who underwent cystectomy, stratifying patients without and with recurrence. In the group of patients with recurrence, more patients received neoadjuvant chemotherapy, had higher tumour stages in TURB and cystectomy, and had positive nodal status. The median follow-up period was 433 days (Q1: 256, Q3; 847) from diagnosis to the end of follow-up. In the follow-up period, 145 out of 180 patients died. In total, 126 patients (70%) died from BC, and, of all patients who died, 87% died from BC. In the Appendix A is provided, comparing patient groups with single and multiple recurrences. Patient characteristics were similar in the two groups. However, in the single-recurrence group, more patients had robot-assisted surgery compared to the multiple-recurrence group.

### 3.1. Cox Regression

The median time to recurrence was 224 days (95% CI: 23.4–1054). The sites of metastases are illustrated in Table 2. In this study, the most common site of recurrence was local recurrence (44%). The most common distant metastatic site was lung metastasis (34%). In total, 108 patients (60%) had two or more recurrences.

Evaluating overall mortality, patients with ≥2 metastases had significantly increased HR’s compared to patients with a single site of recurrence in both crude (HR = 1.59 (95% CI: 1.13–2.25, *p* = 0.01)) as well as adjusted analyses (HR 1.63 (95% CI: 1.15–2.33, *p* = 0.01)). Furthermore, patients with distant metastases had increased HRs compared to patients with only lymph node metastases in crude (HR = 1.59 (95% CI: 0.99–2.56, *p* = 0.06)) as well as adjusted analyses (HR = 1.73 (95% CI: 1.06–2.84, *p* = 0.03)). When dividing patients into subgroups, such as bone metastases, had significantly increased HRs compared to patients with lymph node metastases and local recurrence in both crude (HR = 2.12 (95% CI: 1.06–4.23, *p* = 0.034)) as well as adjusted analyses (HR = 2.08 (95% CI: 1.02–4.24, *p* = 0.045)). We did not find significantly increased HRs in patients with lung or liver metastases compared to patients with lymph node metastases and local recurrence. Furthermore, in single-metastasis-analysis patients, those with bone metastasis had significantly increased HRs compared to patients with lymph node metastases in both crude (HR = 4.25 (95% CI: 1.28–14.1, *p* = 0.02)) as well as adjusted analyses (HR = 3.93 (95% CI: 1.16–13.3, *p* = 0.03)). All HRs are presented in Table 3.

### 3.2. Kaplain Meier Analysis

Overall survival was estimated in the total cohort and in the single recurrence cohort analysis as Kaplan–Meier plots (Figure 1 and Figure 2). There was a statistically significant difference in overall survival when comparing patients with single recurrence and patients with ≥2 metastases (*p* = 0.01). This is presented in Figure 1a. Furthermore, there is a tendency that patients with bone metastases and other metastases or only bone metastases have lower overall survival compared to patients with other metastases, lymph node metastases, and local recurrence. However, there was no significant difference comparing the groups (*p* = 0.05).

Furthermore, there was a statistically difference in overall survival comparing lymph node metastasis, lung metastasis, and other organ metastases in a single recurrence analysis (HR = 0.01). The overall survival was lower in patients with other metastases and lymph node metastases compared to patients with lung metastases in the single-recurrence analysis. The difference in overall survival was significant (*p* = 0.01).

## 4. Discussion

This present study found associations between the location of metastasis and survival probability and additionally found a lower overall survival rate when comparing patients with ≥2 metastases to patients with a single site of recurrence. Corroborating this finding, several previous studies have shown that patients with more disseminated disease have a worse prognosis than patients with only local disease [12,13]. This study found a significantly increased hazard comparing patients with both lymph node and distant metastases to only lymph node recurrence in crude analyses. However, in the adjusted analysis, the association was significant in both patients with distant and lymph node metastases and patients with distant metastases groups compared to patients with only lymph node recurrence. When analysing only a single site of recurrence, a significantly increased hazard was found when comparing other organ metastases than lung metastases to lymph node metastasis. This study also found a statistically significant difference in overall survival probability in the following groups: lymph node metastasis, other metastases, and bone metastasis (*p* = 0.04). Unfortunately, this analysis was not feasible for abdominal metastases (defined by metastases in the liver, carcinomatosis, adrenal gland, and intestine), as this present study population was too small, according to our Schoenfeld residuals calculations.

Other studies have investigated the impact of tumour stage on survival [4,13,14]. Yafi et al.’s study [4] found an increased risk of recurrence associated with higher tumour stage and nodal status. They concluded that varied follow-up protocols would be beneficial. Likewise, this present study found a poorer outcome associated with having ≥2 metastases.

The impact that the location of metastases has on survival has only been sparsely studied. A study of 19 patients found a decreased disease-specific survival in patients with distant recurrence compared to patients with local recurrence, who were treated with curative intentions [5]. Furthermore, the study by Moschini et al. [13] estimated cancer-specific mortality-free survival rates, showing that patients with distant metastases had increased risk. These results resemble both the crude and adjusted analysis in the present study.

When Shou et al. [9] analysed patients with only a single site of recurrence in stage IV BC, they found an association between liver metastasis and a worse survival outcome than other distant metastases. The patients in the study population differed from this present study, as they had not undergone cystectomy and had more advanced disease compared to the present study. Still, this present study finds the same tendency with an increased HR for patients with liver metastases. Another study found a lower disease-specific survival in patients with symptomatic recurrence [5]. In this sense, it is possible that some metastasis locations are more likely to give patients symptoms, and thereby it might be detected earlier, improving the prognosis. On the contrary, if metastases present with symptoms, the likelihood of more advanced disease increases, decreasing the survival probability [12,13]. A study showed that BC patients with symptoms at follow-up were associated with poorer post-recurrence disease-specific survival [15].

In this present study, we investigated overall survival in patients with metastases post cystectomy. In the Shou et al. study [9], they investigated patients with stage IV BC. In this present study, we wanted to investigate metastatic pattern and the impact on survival, a topic only sparsely investigated. In our study, we found that patients with bone metastases had poorer prognosis and that patients with lung metastases had better prognosis compared to patients with lymph node metastases. This replicates the Shou et al. study, as they also found higher mean survival in patients with lung metastases compared to lower mean survival in patients with bone metastases.

In other cancer diseases, studies have found associations between the site of recurrence and prognosis. In the Wang et al.’s study [8] investigating breast cancer, they found that patients with brain metastases had the worst prognosis in cancer-specific survival in all breast cancer subtypes. From these results, it would make sense that the location of metastases also has an impact on patients’ prognosis in bladder cancer.

In an autopsy study from 1999, 47% percent of patients had both lymph node and distant metastases [16]. In this present study, only 26% had both lymph node and distant recurrence. This could be due to the development of technologies such as PET-CT in diagnosis, better surgical techniques, and more developed knowledge regarding the importance of the removal of lymph nodes during surgery. Furthermore, 12% had only lymph node metastasis in the autopsy report. In this present study, 10% of all patients with recurrence had only lymph node recurrence. This could support the hypothesis, as, in our study, we see fewer lymph node metastases (51%) in general compared to the autopsy report (96% of all patients with recurrence).

This present study is a retrospective cohort study. The risk of information bias is present, as data were collected by different investigators which could have evaluated the data differently. However, we do believe that this effect might be minimal due to a small number of investigators, and it will most likely be differentiated misclassification, which will lead to bias towards the null hypothesis. This present study is also a single-centred study, which decreases its external validity. Further studies are needed to confirm external validity. Additionally, there is risk of confounding other characteristics than those that were adjusted for. Other possible confounders could be tumour stage, smoking status, and treatment after recurrence. Unfortunately, adjuvant treatments were not registered as explicit enough to allow for meaningful adjustments, and this is thus a limitation. This could be a relevant factor to adjust in future studies. Furthermore, possible confounders could also be caused by known and unknown comorbidities, for example AMI, apoplexy, or other malignancies, which could be relevant to incorporate in future studies. Furthermore, other analytical approaches will be relevant to future studies. Instead of using overall survival as the outcome, analyses of disease-specific survival could be relevant. However, this would increase risk of misclassification. A larger study population would have been preferable, as it would have been interesting to divide patients into subgroups of metastases, for example, lung and liver, bone and liver, etc. Many of the patients in this study had more than one metastasis, but if we divided the patients into subtypes, it would have resulted in small groups, and therefore was not preferable with this study population.

To the best of our knowledge, no previous study has investigated the association between the specific location of recurrence and survival in BC patients diagnosed with recurrence after radical cystectomy. The association is difficult to analyse, as there are many factors affecting patients’ outcomes. However, the location of metastases seems to affect the overall mortality, as the present study, among others, found a higher overall mortality for patients diagnosed with liver and bone metastasis. Further studies are needed to verify the results from the present study and to identify in more detail the impact of metastases location on patients’ prognoses. This information could aid clinicians in choosing a treatment strategy, allowing for the selection between a milder or more aggressive approach based on the site of recurrence.

## 5. Conclusions

To the best of our knowledge, this study is the first to investigate the impact of specific localization of recurrence in BC patients after radical cystectomy. Investigating overall survival, we found a significantly lower survival probability in patients with multiple metastatic sites compared to patients with a single site of recurrence. Furthermore, we observed a lower survival probability for patients who had bone metastases compared to patients with lymph node metastases in our single metastasis analysis.

## Figures and Tables

**Figure 1 cancers-17-00867-f001:**
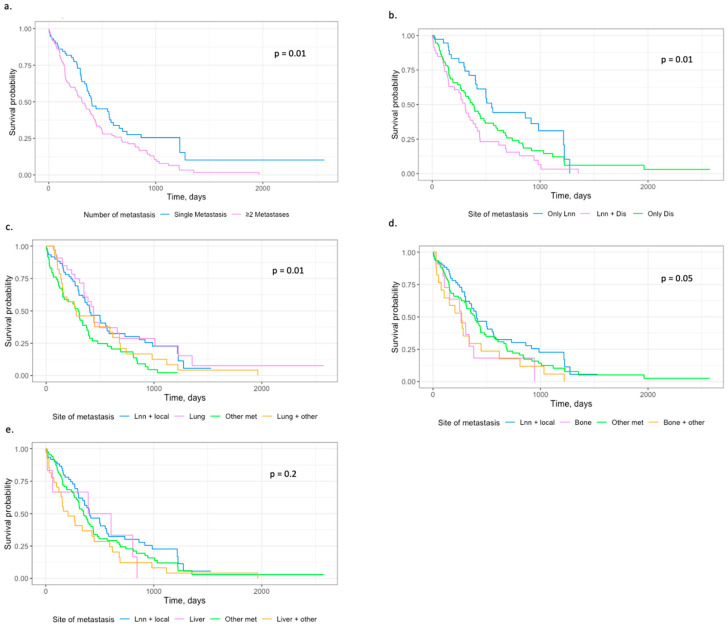
Kaplan–Meier curves. Survival probability as a function of time measured in days. (**a**) Groups divided in single metastasis and ≥2 metastases. (**b**) Groups divided in Lnn: “Only lymph node metastases”, Lnn + Dis: “Lymph node and distant organ metastases”, Only Dis: “Only distant organ metastases”. (**c**–**e**) Groups divided in “Lnn + local”: “Lymph node and local recurrence”, “Other met”: “Other organ metastases” and lung, bone, and liver, respectively—both as single and combined with other organ metastases.

**Figure 2 cancers-17-00867-f002:**
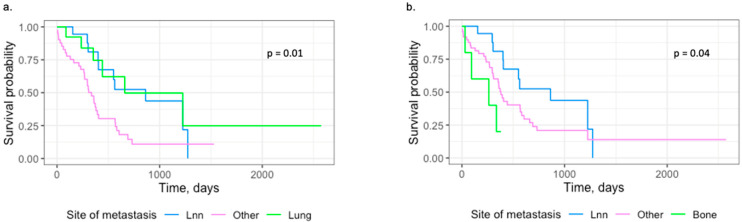
Kaplan–Meier plots on single metastasis analysis. (**a**) Compares survival probability in the following groups: “Lymph node metastasis”, “Other organ metastases”, “Lung metastasis”. (**b**) Compares survival probability in the following groups: “Lymph node metastasis”, “Other organ metastases”, “Bone metastasis”.

**Table 1 cancers-17-00867-t001:** Patient demographics.

Characteristic	N = 180 ^1^
Sex	
Male	124 (69%)
Female	56 (31%)
Age	73 (66, 78)
BMI	
<25	65 (36%)
25–29.9	79 (44%)
>30	36 (20%)
ASA Classification	
1	13 (7.2%)
2	105 (58%)
3	62 (34%)
Smoking Status	
Never	33 (18%)
Ceased > 5 Years Ago	75 (42%)
Ceased < 5 Years Ago	16 (8.9%)
Smoker	51 (28%)
Unknown	5 (2.8%)
Neoadjuvant Chemotherapy	
Yes	67 (37%)
No	113 (63%)
BCG Treatment	
Yes	16 (8.9%)
No	164 (91%)
Robot Assisted	80 (44%)
Urinary Conduit	
Ileal Conduit	170 (94%)
Other	10 (5.6%)
Tumour Stage (TURB)	
Unknown	<4 (<2.2%)
Ta, CIS or T1	44 (24%)
T2	132 (73%)
≥T3	<4 (<2.2%)
Tumour Stage (Cystectomy)	
Unkown	<4 (<2.2%)
T0	29 (16%)
Ta	4 (2.2%)
CIS	4 (2.2%)
T1	9 (5.0%)
T2	16 (8.9%)
≥T3	116 (64%)
Lymph Nodes Removed	24 (15, 34)
Nodal Status	
Positive	60 (33%)
Negative	120 (67%)

^1^ n (%); median (IQR).

**Table 2 cancers-17-00867-t002:** Number and incidence of metastases. Including the total cohort and subgrouped cohort according to ≥2 metastases or single metastases.

Characteristic	All	Stratified		
	N = 180 ^1^	Single Site of Recurrence, N = 72 ^1^	≥2 Metastases, N = 108 ^1^	*p*-Value ^2^
Local Recurrence	79 (44%)	23 (32%)	56 (52%)	0.008
Local Lymph Node	34 (19%)	11 (15%)	23 (21%)	0.3
Distant Lymph Node	57 (32%)	7 (9.7%)	50 (46%)	<0.001
Liver	33 (18%)	<4 (<5.6%)	30 (28%)	<0.001
Lung	61 (34%)	13 (18%)	48 (44%)	<0.001
CNS	7 (3.9%)	<4 (<5.6%)	5 (4.6%)	0.7
Skin	10 (5.6%)	<4 (0%)	10 (9.3%)	0.006
Bone	28 (16%)	5 (6.9%)	23 (21%)	0.009
Peritoneal Carcinomatosis	16 (8.9%)	<4 (<5.6%)	14 (13%)	0.019
Adrenal Gland	4 (2.2%)	<4 (<5.6%)	<4 (<3.7%)	0.7
Intestine and Other Internal Organs	7 (3.9%)	<4 (<5.6%)	7 (6.5%)	0.043
Soft Tissue	4 (2.2%)	<4 (<5.6%)	<4 (3.7%)	>0.9
Penile	<4 (<2.2%)	<4 (<5.6%)	<4 (<3.7%)	0.5
Mediastinum	<4 (<2.2%)	<4 (<5.6%)	<4 (<3.7%)	0.5
ctDNA *	4 (2.2%)	<4 (<5.6%)	<4 (<3.7%)	0.3

^1^ n (%); ^2^ Pearson’s Chi-squared test; Fisher’s exact test. * Participated in a research protocol measuring circulating tumour DNA (ctDNA).

**Table 3 cancers-17-00867-t003:** Crude and adjusted HRs for overall mortality for both the total cohort (n = 180) and for patients with only a single metastasis (n = 72).

	Crude HR			Adjusted HR		
	HR	95% CI	*p*-Value	HR	95% CI	*p*-Value
Total cohort						
Lymph node and distant metastases *	2.25	1.36–3.75	0.002	1.84	1.06–3.19	0.03
Distant organ metastases *	1.59	0.99–2.56	0.06	1.73	1.06–2.84	0.03
≥2 metastases **	1.59	1.13–2.25	0.01	1.63	1.15–2.33	0.01
Lung ***						
Lung	0.88	0.53–1.46	0.6	0.88	0.52–1.50	0.6
Lung and other organ metastases	1.37	0.84–2.24	0.2	1.16	0.67–2.02	0.6
Other organ metastases (not lung)	1.84	1.23–2.76	0.003	1.91	1.23–2.98	0.004
Bone ***						
Bone	2.12	1.06–4.23	0.034	2.08	1.02–4.24	0.045
Bone and other metastases	1.88	1.03–3.30	0.027	1.48	0.80–2.73	0.2
Other organ metastases (not bone)	1.24	0.85–1.81	0.3	1.24	0.82–1.87	0.3
Liver ***						
Liver	1.44	0.61–3.39	0.4	1.88	0.74–4.76	0.2
Liver and other organ metastases	1.74	1.07–2.84	0.026	1.55	0.88–2.72	0.13
Other organ metastases (not liver)	1.27	0.87–1.86	0.2	1.27	0.84–1.91	0.3
Single metastasis ****						
Lung	0.83	0.30–2.29	0.7	0.82	0.28–2.42	0.7
Other organ metastases than lung	2.42	1.18–4.95	0.02	2.34	1.11–4.93	0.03
Bone	4.25	1.28–14.1	0.02	3.93	1.16–13.3	0.03
Other organ metastases than bone	1.75	0.86–3.54	0.12	1.76	0.82–3.76	0.15

* Reference group: Only lymph node metastases. ** Reference group: Single metastasis. *** Reference group: Lymph node metastases and local recurrence. **** Reference group: Lymph node metastases.

## Data Availability

The datasets presented in this article are not readily available due to technical limitations. Requests to access the datasets should be directed to Jørgen Bjerggaard Jensen.

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
