# Peer review of "The Impact of the Localization of Metastasis in Bladder Cancer Patients with Recurrence After Cystectomy"

_cancers, 2025, doi:10.3390/cancers17050867_

Round 1
Reviewer 1 Report (Previous Reviewer 2)
Comments and Suggestions for Authors
In general, the authors have been responsive to the reviewer's comment. However, some suggestions for additional analyses were not satisfied. It is important to address the following:
1. Given that the majority of cancer patients died during follow-up it is important to evaluate not only overall mortality but also cancer-specific mortality using competing risk analysis. If most deaths were due to bladder cancer given patients' clinical characteristics then please add a sentence denoting what proportion of deaths were due to cancer.
2. The authors state that "Between 2015-2021, 666 patients underwent radical cystectomy, of which 180 (27%) had recurrence and were included in the study" it will be helpful if they provide a supplemental table to compare characteristics of Table 1 between the two groups.
3. How many of the 180 patients have one recurrence and how many have two or more? Were their characteristics comparable?
4. How do the results of this study compare with other previously published studies?
Author Response
Comment 1: Given that the majority of cancer patients died during follow-up it is important to evaluate not only overall mortality but also cancer-specific mortality using competing risk analysis. If most deaths were due to bladder cancer given patients' clinical characteristics then please add a sentence denoting what proportion of deaths were due to cancer.
Thank you for the comment. We have made a table for you stratified data on patients cause of mortality. 126 patients died because of BC, so most patients died because of cancer. We have added a sentence in line 94-95.
Comment 2: The authors state that "Between 2015-2021, 666 patients underwent radical cystectomy, of which 180 (27%) had recurrence and were included in the study" it will be helpful if they provide a supplemental table to compare characteristics of Table 1 between the two groups.
Thank you, we have made a table and named it “Table All cystectomies in 2015-2021”. We have added the table in a supplementary attachment. Furthermore, a short description is added in line 90-92.
Patients with higher TUR-B and with positive lymph node in cystectomy were more frequent in patients with recurrence. However, this is something that has been researched before and not the subject of interest in our study.
Furthermore, of the 666 patients two were excluded as they underwent cystectomy do to other reasons than bladder cancer that is why there is only 664 in the table.
Comment 3: How many of the 180 patients have one recurrence and how many have two or more? Were their characteristics comparable?
Thank you, we have clarified this in the manuscript. 108 patients had 2 or more recurrences. We have added this in line 102-103.
We have provided a table Single vs. Multiple Recurrence like you asked for. As you can see the two groups are not significantly different in almost all the parameters.
Comment 4: How do the results of this study compare with other previously published studies?
Thank you for your comment. As we also write in the manuscript not much research has been made on this specific subject. We have compared our study with previous literature in these sections: (line 155-159 on multiple recurrence), (line 171-195 on site of recurrence).

Round 2
Reviewer 1 Report (Previous Reviewer 2)
Comments and Suggestions for Authors
Authors have provided all relevant requested information. However, it will help if in their results they comment elaborate a bit regarding the comparison of characteristics between the two groups of patients presented in Supplementary Table 1... e.g. most characteristics were similar with exception of tumor stage and nodal status.
Same comment goes also for Supplementary table 2 which is more important for their paper.
In the methods they should also state how was cause of death determined.
Author Response
Comment 1: Authors have provided all relevant requested information. However, it will help if in their results they comment elaborate a bit regarding the comparison of characteristics between the two groups of patients presented in Supplementary Table 1... e.g. most characteristics were similar with exception of tumor stage and nodal status.
Thank you for the acknowledgement of our work. We have added a comparison in line 93-95.
Comment 2: Same comment goes also for Supplementary table 2 which is more important for their paper.
Thank you for the comment. We have clarified this on line 98-101
Comment 3: In the methods they should also state how was cause of death determined.
Thank you for your comment. We have clarified this on line 73-74.
This manuscript is a resubmission of an earlier submission. The following is a list of the peer review reports and author responses from that submission.
Round 1
Reviewer 1 Report
Comments and Suggestions for Authors
The article under review is a retrospective cohort study aiming to analyze the effects of distant metastases location on the prognosis of patients that underwent radical cystectomy for muscle invasive bladder cancer. It is one of the first papers that that evaluates this association and as such, this article scores great when it comes to novelty.
The study had a large group of patients included (666 patients with cystectomy from 2015 to 2021, from which 180 with recurrence were included ) with a very long follow-up period (433 days), having such a large database to work with is always challenging, showing great interest from the research team. The presentation of the material and methods is straight forward showing great implication from the team in order to obtain statistically relevant results.
Results are clear and concise presenting the relevant information to the study which showed an association between the number of metastases and a worse prognosis, and patients with bone metastases showing the worse prognosis of all metastatic sites.
Discussions are clear and are relevant to the subject of the study. As the authors pointed out by themselves, being a limit of this research is the fact that it is a retrospective study which brings a risk for bias.
Conclusions are concise and present they answer the study aim that was presented.
Author Response
Comment 1: The article under review is a retrospective cohort study aiming to analyze the effects of distant metastases location on the prognosis of patients that underwent radical cystectomy for muscle invasive bladder cancer. It is one of the first papers that that evaluates this association and as such, this article scores great when it comes to novelty. The study had a large group of patients included (666 patients with cystectomy from 2015 to 2021, from which 180 with recurrence were included ) with a very long follow-up period (433 days), having such a large database to work with is always challenging, showing great interest from the research team. The presentation of the material and methods is straight forward showing great implication from the team in order to obtain statistically relevant results. Results are clear and concise presenting the relevant information to the study which showed an association between the number of metastases and a worse prognosis, and patients with bone metastases showing the worse prognosis of all metastatic sites. Discussions are clear and are relevant to the subject of the study. As the authors pointed out by themselves, being a limit of this research is the fact that it is a retrospective study which brings a risk for bias. Conclusions are concise and present they answer the study aim that was presented.
Thank you for your time and comments. We agree with the statements.
Reviewer 2 Report
Comments and Suggestions for Authors
In this paper the authors aimed to investigate whether the specific localization of recurrence and the number of metastases were associated with overall survival among bladder cancer patients from a single institution in Denmark.
The authors state that they identified 666 patients who underwent radical cystectomy between 2015 and 2021, of which 180 (27%) had recurrence and were included in the study. Were the patient included diagnosed with muscle invasive bladder cancer? It might be helpful to have a flowchart with inclusion and exclusion criteria and respective sample sizes.
Why weren't the other patients included in the study? What was the overall survival among patients who didn't have a recurrence? It might be helpful to compare demographic and clinical characteristics of patients who were included vs those excluded in the study, as well as overall survival.
The authors focus only on overall survival, but it would be helpful to consider also cancer-specific survival as the most interesting outcome of interest. Can the authors run a competing risk analyses for BC-specific survival to examine their hypothesis?
How many patients died during the follow-up period? Were the patients with various metastasis included in various models in Table 3? If so were the models adjusted for multiple comparisons?
In the discussion the author should mention potential for selection bias and survival bias if their analyses are focused only on patients with more advanced cancer.
Author Response
Comment 1: Were the patient included diagnosed with muscle invasive bladder cancer?
Patientdata is presented in table 1. As described most patients had muscle invasive bladder cancer. According to Danish guidelines, radical cystectomy is offered for patients with MIBC, T1b and recurrence CIS which reflect the included population.
Comment 2: Why weren't the other patients included in the study? What was the overall survival among patients who didn't have a recurrence? It might be helpful to compare demographic and clinical characteristics of patients who were included vs those excluded in the study, as well as overall survival.
This is an interesting study as well. The resources was limited and therefor also the scope of the study was limited. The patients of interest in our study was patients with bladder cancer that experienced recurrence. As this subject has only been studied sparsely before.
Comment 3: The authors focus only on overall survival, but it would be helpful to consider also cancer-specific survival as the most interesting outcome of interest. Can the authors run a competing risk analyses for BC-specific survival to examine their hypothesis?
We chose overall survival because we did not want to increase risk of information bias as described (line 225-227). That is the reason we chose overall survival.
Comment 4: How many patients died during the follow-up period? Were the patients with various metastasis included in various models in Table 3? If so were the models adjusted for multiple comparisons?
145 out of the 180 patients died (line 91). The patients in table 3 were only compared in the defined groups. So HR's should be interpreted in specific groups comparing patients with the regarding reference groups as described.
Comment 5: In the discussion the author should mention potential for selection bias and survival bias if their analyses are focused only on patients with more advanced cancer.
This is important to underline, thank you. We have already mentioned risk for selection bias. However, our analyses are not focused on patients with more advanced disease at time of cystectomy. The patients we investigated had recurrence to be included in the study and therefore had an advanced disease, but not from time of diagnosis.
Reviewer 3 Report
Comments and Suggestions for Authors
Review for the article
“The impact of localization of metastasis in bladder cancer patients with recurrence after cystectomy.”
Summary
The cancers-3310986 is an interesting topic. The authors presented a single-center retrospective study, investigating the the prognostic implications of metastatic anatomical location in patients with recurrent bladder cancer after radical cystectomy. The manuscript is solid and well-written.
Abstract
The abstract is short and complete, allowing the reader to understand the topic and presenting adequately the necessary information about the study.
Introduction
- The information provided in this section is valuable for the comprehension of the manuscript. The authors provide data about bladder cancer (BC) and the frequency of metastatic disease. The scientific background regarding the possible significance of the localization of metastasis is also presented.
- The objective of the study is clearly mentioned in the last paragraph, while the scientific gap regarding the aim of the study is underlined.
Methods
- The study design is well explained. The number of patients included (which is adequate) could be mentioned in this section.
- The authors inform the reader about inclusion criteria, while all the necessary definitions are correctly presented in this section. In addition, the data collected are mentioned in this section.
- Disease recurrence was evaluated radiologically, and histologically (when possible), while ctDNA was also used.
- The intensity of the follow-up of the participants is also mentioned.
Results
- The results are presented in an extensive way. The authors underline that bone metastasis seems to have the worst prognosis when compared to other single metastasis sites.
- The tables and figures are really helpful and necessary for the completion of the authors' work.
Discussion
- The discussion is of good quality and includes updated data.
- The authors analyzed the existent citations, using suitable references in order to complete and compare their findings.
- The authors inform extensively the reader about the study limitations, while they correctly underline the importance of more studies (maybe prospective and multicenter) for the generalization of the aforementioned results.
Conclusion
From the presented data, the conclusion is complete and represents the work that the authors did.
Author Response
Comment 1: Summary The cancers-3310986 is an interesting topic. The authors presented a single-center retrospective study, investigating the the prognostic implications of metastatic anatomical location in patients with recurrent bladder cancer after radical cystectomy. The manuscript is solid and well-written.
Abstract The abstract is short and complete, allowing the reader to understand the topic and presenting adequately the necessary information about the study. Introduction
- The information provided in this section is valuable for the comprehension of the manuscript. The authors provide data about bladder cancer (BC) and the frequency of metastatic disease. The scientific background regarding the possible significance of the localization of metastasis is also presented.
- The objective of the study is clearly mentioned in the last paragraph, while the scientific gap regarding the aim of the study is underlined.
Methods
- The study design is well explained. The number of patients included (which is adequate) could be mentioned in this section.
- The authors inform the reader about inclusion criteria, while all the necessary definitions are correctly presented in this section. In addition, the data collected are mentioned in this section.
- Disease recurrence was evaluated radiologically, and histologically (when possible), while ctDNA was also used.
- The intensity of the follow-up of the participants is also mentioned.
Results
- The results are presented in an extensive way. The authors underline that bone metastasis seems to have the worst prognosis when compared to other single metastasis sites.
- The tables and figures are really helpful and necessary for the completion of the authors' work.
Discussion
- The discussion is of good quality and includes updated data.
- The authors analyzed the existent citations, using suitable references in order to complete and compare their findings.
- The authors inform extensively the reader about the study limitations, while they correctly underline the importance of more studies (maybe prospective and multicenter) for the generalization of the aforementioned results.
Conclusion From the presented data, the conclusion is complete and represents the work that the authors did.
Thank you for your time and comments. We agree with the statement.